# Research Update on the Impact of Lactic Acid Bacteria on the Substance Metabolism, Flavor, and Quality Characteristics of Fermented Meat Products

**DOI:** 10.3390/foods11142090

**Published:** 2022-07-14

**Authors:** Yi Wang, Jun Han, Daixun Wang, Fang Gao, Kaiping Zhang, Jianjun Tian, Ye Jin

**Affiliations:** 1College of Food Science and Engineering, Inner Mongolia Agricultural University, Hohhot 010018, China; plwangyi99@163.com (Y.W.); hanj0928@126.com (J.H.); wungdaixun@163.com (D.W.); gf2899760971@163.com (F.G.); jinyeyc@sohu.com (Y.J.); 2Ministry of Agriculture and Rural Affairs Integrative Research Base of Beef and Lamb Processing Technology, Hohhot 010018, China; 3Department of Cooking & Food Processing, Inner Mongolia Business and Trade Vocational College, Hohhot 010070, China; zhangkaiping2005@163.com

**Keywords:** fermented meat products, substance metabolism, functional characteristics

## Abstract

This paper reviews the effects of domestic and foreign influences on the substance metabolism pathways and the flavor and flora of LAB in fermented meat products to provide a new theoretical basis for developing new products for the industrial application of lactic acid bacteria (LAB) in fermented meat products. LAB are extensively used among commonly fermented ingredients, such as fermented meat products and yogurt. As fermenting agents, LAB metabolize proteins, lipids, and glycogen in meat products through their enzyme system, which affects the tricarboxylic acid cycle, fatty acid metabolism, amino acid decomposition, and other metabolic processes, and decompose biological macromolecules into small molecules, adding a special flavor with a certain functionality to the final product. Metabolites of LAB in the fermentation process also exert nitrite degradation, as well as antibacterial and antioxidant functions, which improve the physical and chemical qualities of fermented meat products. While fermenting meat products, LAB not only add unique flavor substances to the products, but also improve the safety profile of fermented foods.

## 1. Introduction

Fermented meat products are obtained by fermenting raw meat with specific microorganisms or enzymes under natural or artificial intervention conditions to induce a series of biochemical reactions and physical changes [1]. Most of the current fermented meat products are fermented in a natural environment, and the problems with their use mainly include an unclear source of fermentation strains, uncontrollable microbial flora, easy contamination by harmful pathogenic bacteria, and other food safety problems [2,3].

As probiotics offering multiple functions, LAB improve the color, flavor, and pH of meat products and inhibit the growth of harmful microorganisms during the fermentation process [4]. They also decompose large molecules of protein, fat, and carbohydrates in the meat to generate a large number of aromatic substances such as alcohols, aldehydes, acids, and esters, as well as amino acid and small molecule peptides, which can be easily absorbed by the body and produce lactic acid, lactic acid bacteriocins, and other antibacterial metabolites to degrade nitrate in meat products [5]. LAB are used in the fermentation of meat products due to their ability to degrade nitrites, provide an antioxidant activity, and inhibit the growth of pathogenic bacteria [6]. Meanwhile, they reduce the moisture content in the meat, which effectively improves product safety and extends shelf life [7]. This paper reviews the current status of research on the effects of LAB on the substance metabolism, flavor, and flora of fermented meat products.

## 2. Effects of LAB on Substance Metabolism and the Flavor of Fermented Meat Products

LAB offer a unique enzyme system during the fermentation process, as they not only react with glucose to produce acids, but also produce protease and lipase, which break down the proteins, fats, and carbohydrates in the food into small molecules, and some metabolites react with each other to produce flavorful substances. LAB are the core microorganisms involved in the fermentation process of meat products [8], and the common dominant LAB include *Lactobacillus* spp., represented by *L. plantarum*, *L. sake*, *L. paracasei*, and *L. fermentum* [9,10,11,12,13,14]; *Streptococcus* spp., represented by *S. pentosus* [15,16,17,18]; *Enterococcus* spp., represented by *E. faecalis* [19,20,21,22,23]; and *Lactobacillus lucidus* spp. and *Lactococcus* spp. The flavors produced mainly arise from the hydrolysis of proteins, fats, and carbohydrates by enzymes, and the precursors of aromatic compounds are mainly produced by free-fatty acids and amino acids [24,25]. Wen et al. [26] found that inoculation with *P. acidilactici* BP2 enhanced the flavor profile and acceptability of beef jerky. The production of flavor compounds in fermented sausages are mainly based on microbial metabolism and chemical reactions. The microbial metabolism includes carbohydrate fermentation, amino acid degradation, fatty acid β-oxidation, and staphylococcal esterase enzymatic digestion. The chemical reactions include auto-oxidation of fats, the Maillard reaction, and the Strecker degradation reaction, which produces smaller molecules of diacetyl, acetic acid, and acetone aldehyde.

### 2.1. Effects of LAB on Protein Metabolism and Flavor

Lactic acid is produced during the fermentation of LAB, giving the product a special flavor, and its enzyme system produces esterase, protease, and peroxidase; these enzymes and endogenous enzymes in the meat products synergize, inducing biochemical changes in the protein and fat of raw meat, thereby producing small molecules of amino acids, esters, peptides, and short-chain fatty acids and other small molecules [27]. Presently, most raw meat used in processed meat products consists of skeletal muscles, in which the proteins are mainly myogenic fibrin, sarcoplasmic protein, and matrix protein, and LAB can hydrolyze the above-mentioned muscle proteins into oligopeptides, while the oligopeptides convert the free small peptides and amino acids into α-keto acids through transamination reactions, which in turn generate the corresponding amino acids through the ammonia addition reaction; α-keto acids are also precursors of alcohols with fruit flavors and other aromatic substances [28]. LAB can biochemically break down proteins in the meat and influence the gel characteristics of meat products through acid production, which in turn affects their physicochemical properties.

#### 2.1.1. Effects of LAB on Protein Metabolism and on the Flavor and Physicochemical Properties of Meat Products

As the fermentation time of meat increases, structurally, the secondary and tertiary structures of the proteins are changed, mainly in the form of a decrease in the alpha-helix and an increase in the beta-fold. The metabolic pathways of LAB on proteins in fermented meat products and the flavor substances produced are depicted in Figure 1. When LAB utilize myofibrillar proteins and sarcoplasmic proteins, they are initiated by cell-envelope proteinase, and they break down the proteins into polypeptides, which are then degraded into oligopeptides and amino acids through the action of peptidase, and enter the cell through the transport channels of oligopeptides and amino acids; the oligopeptides are broken down into short peptides and amino acids by the action of endopeptidases in the cells [29]. Short peptides and amino acids are converted from asparagine to alanine, which has a fresh sweet taste, through the action of histoproteinase, and cysteine and serine are converted into pyruvate, which is the core substance of the tricarboxylic acid cycle; branched-chain amino acids are converted into α-keto acids and generate aldehydes such as 2-methyl butyraldehyde and 3-methyl butyraldehyde through the action of keto acid dehydrogenase, aldehydes generate alcohols and acids through the action of alcohol dehydrogenase aldehyde dehydrogenase, and alcohols and acids generate aromatic esters through the action of esterase. In particular, leucine, under the action of transaminase and dehydrogenase, produces 3-methyl butanol with a whiskey flavor and maltiness and 3-methyl butyric acid with a sour flavor. Butyric acid is a short-chain fatty acid produced through the fermentation of dietary fiber under the action of intestinal microorganisms, which exerts the functions of providing energy for intestinal epithelial cells; strengthening the intestinal mucosal barrier; and regulating immunity, inflammation, and oxidation [30].

Chen et al. [31] found that the abundance values of various flavor substances increased with protein hydrolysis when the fermentation time reached 9 days, in addition to which different proportions of salt also affected the flavor, pH, and moisture content of the fermented sausages. Fadda et al. [32] and Sanz et al. [33] reported that *L. plantarum*’s enzyme system can break down myogenic fibronectin and sarcoplasmic proteins into small active peptides and free the amino acids through the action of proteases. Free amino acids are mostly used as a source of flavor substances and can also produce various flavor components under the action of different enzymes; for example, valine has the aroma of malt, glycine and threonine are sources of sweet flavors; glutamic acid is a source of fresh and savory flavors, and aminobutyric acid produced by glutamic acid under the action of decarboxylase can also prevent diabetes [34]. The accumulation of various peptides and amino acids facilitates the enhancement of meat aroma and flavor.

In general, proteins undergo oxidative deamination and decarboxylation reactions under the action of a series of enzymes to eventually produce flavor precursors, such as aldehydes, alcohols, and aromatic substances.

#### 2.1.2. Improvement in the Tenderness and Color of Meat Products by LAB

LAB, as a natural source of fermenting agents in the fermentation process of meat products, affect physical properties such as color and texture. LAB produce acids during the fermentation process to lower the pH, while the pH affects the water-binding ability and water-holding capacity of myogenic fibrin. When the pH is lower than the isoelectric point of myogenic fibrin, the hardness and chewiness of fermented meat products increase. After the raw meat is cured with salt, the salt-soluble proteins are dissolved, and these dissolved proteins are denatured through the action of lactic acid to form a gel-like structure [24].

Liu et al. [35] reported the results of the dynamic rheological properties of myosin, wherein the mobility of protein molecules increases when the pH is greater than the isoelectric point of myosin, leading to an increase in the viscoelasticity of the myofibril mixture. It has been reported [36] that fermented meat inoculated with LAB develops better textural properties, such as hardness, when compared with natural meat during the ripening process. Zhu et al. [37] reported that *L. plantarum* could reduce nitrate and nitrite and reduce the hazards of biogenic amines; this strain could improve the color and gel characteristics of sausage during fermenting. *L. plantarum* improves the gelation properties of sausages during fermentation mainly by promoting protein unhelixation, hydrophobic interaction, β-folding, and hydrogen bonding forces between protein and water, and by producing nitroso, which reacts with myoglobin in the meat to produce bright red nitrosomyoglobin (MbO_2_). Gou et al. [38] found that nitrate reductase and nitric oxide synthased in LAB are involved in the production of NO, which is associated with the formation of MbNO_2_. Salama et al. [39] applied the red pigment extracted from *Trichoderma purpureum* to meat products and found that the pigment not only gave the meat products a good color even after steaming and frying, but also extended the shelf life of the food. Xu and Zhu [40] reported that *L. plantarum* could reduce the sausage pH, accelerate the acidification and gelatinization process, and add good color to the meat products and reduce the content of nitrite residues in the sausage, as well as prevent fat oxidation, protein decomposition, and myoglobin oxidation, and increase the content of free amino acids.

### 2.2. Effects of LAB on Fat Metabolism and Flavor

Lipolysis is usually considered to be related to microbial lipase activity, while lipid hydrolysis and protein hydrolysis are conventionally considered to play a central role in aroma formation. Fat, as an important source of fermentation flavor, is capable of producing free-fatty acids and glycerides, while polyunsaturated fatty acids are also capable of being oxidized by microorganisms to produce aldehydes and ketones for flavor development [36,41,42].

#### 2.2.1. Fatty Acids and Their Oxidation Products

During the fermentation process of meat products, fat produces free-fatty acids and aldehyde and ketone flavor substances under the action of lipase, and its metabolic products can provide a carbon source for LAB to further produce other flavoring substances. The metabolism of flavor substances through LAB is depicted in Figure 2.

Lipase can break down the lipid substrates triglycerides and phospholipids into free-fatty acids, esterel coenzyme-A and alcohols produce esters of flavor substances through the action of esterase, and alcohols and acids produce esters through the action of esterase. Unsaturated fatty acids produce aldehydes when subjected to the action of peroxidase and produce alcohols and acids through the action of dehydrogenase and hydrogenase; saturated fatty acids are subjected to the action of thiolase, β-ketoacid decarboxylase, and reductase to produce a flavor substance—secondary alcohols. The oxidation of volatile compounds, such as unsaturated fatty acids including oleic and linoleic acids, resulting from lipid oxidation, is considered an important flavor compound in fermented sausages. In addition, valeraldehyde with almond and malt aroma, octanal with fatty and lemon flavor, and heptanal with fatty and citrus flavor are other such compounds. Lactic acid, as a fermentation product, can improve the utilization of calcium, phosphorus, and iron [43]. Bozkurt and Erkmen [44] found that the Thiobarbituric Acid Reactive Substance values of sausages inoculated with LAB were significantly lower than those of uninoculated sausages during the late storage period, indicating that it could inhibit the occurrence of excessive fat oxidation. Uppada et al. [45] reported that *L. plantarum* could be a source of lipase that could degrade fat in meat within 72 h and was capable of esterification reactions to produce short-chain fatty acids. Chen et al. [41] inoculated *Lactobacillus pentosus*, *Lactobacillus campestris*, *Lactobacillus sake*, and *Staphylococcus xylosus* into Harbin dry sausages, and showed that LAB could promote lipid hydrolysis, inhibit lipid autoxidation, and improve the flavor of fermented sausages. Wang et al. [46] found that LAB fermentation of cured fish not only accelerated the degree of fat hydrolysis in the fish, but also significantly increased the content of functional eicosapentaenoic acid (EPA) and docosahexenoic acid (DHA) omega-3 unsaturated fatty acids.

In summary, when LAB are involved in fat metabolism, they can provide carbon sources for it, as well as produce more flavor substances and functional substances.

#### 2.2.2. Potential of LAB in Cholesterol Degradation

As an important component of human cell membranes, cholesterol is an indispensable substance for the body [47]; however, when it is present in excessive amounts in the body, it can cause blockages of the blood vessels and poor blood flow, thereby affecting lipid metabolism and causing cardiovascular diseases and atherosclerosis [48]. LAB regulate the absorption of cholesterol in the body, and the most important mechanism of cholesterol degradation in the body is the conversion of cholesterol to bile acids. After entering the body, LAB adhere to the intestinal lining and interact with the intestinal flora to convert cholesterol into bile acids by influencing the metabolism of intestinal microorganisms, which are then excreted from the body [49]. Table 1 shows some of the recent studies on the degradation of cholesterol by LAB, with the highest degradation rate reported as being 79.00%.

Cholesterol metabolism can be classified into several processes, including cholesterol synthesis, catabolism, and absorption and transportation. LAB phosphorylation of adenosine 5’-monophosphate activated protein kinase (AMPK), the key enzyme in the endoplasmic reticulum membrane, decreases the gene expression of 3-hydroxy-3-methylglutaryl-CoA reductase (HMGCR). Bile salt hydrolase regulates the expression of cholesterol 7α-hydroxylase (CYP7A1), which in turn restricts the production of bile acids and accelerates the degradation of cholesterol in the liver. This is done by downregulating the functional proteins that specifically mediate cholesterol absorption, Niemann-Pick C1-like (NPC1L1) protein, and upregulating ATP-binding cassette transporter G5(ABCG5) and ATP-binding transporter G8(ABCG8), regulating the sterol-regulatory element-binding proteins, in order to inhibit the low-density lipoprotein receptor pathway and reduce the cholesterol content in the body [59]. LAB mediate cholesterol-related genes: firstly, AMPK phosphorylation affects HMGCR gene expression to limit cholesterol synthesis, the expression of CYP7A1 to affect cholesterol catabolism, the expression of NPC1L1 and ABCG5/8 to mediate the absorption and transport of intestinal cholesterol, and the expression of SREBPs to affect cholesterol uptake and transport in the liver [60].

Liang et al. [61] found that *Lactobacillus lactis* F17 and *L. plantarum* F3-2 reduced the trimethylamine levels in the intestine and decreased the serum levels of trimethylamine and oxidized trimethylamine, while the probiotic preparations improved the lipid profiles through famesoid X receptor (FXR) and CYP7A1 to improve the lipid metabolism as well as promote bile acid and cholesterol metabolism by regulating the genes CYP7A1 and FXR and reducing the triglyceride (TG) and serum total cholesterol levels. In conclusion, the use of LAB for cholesterol degradation is not only safe, but it is also devoid of potential adverse effects such as drug resistance.

### 2.3. Effects of LAB on Carbohydrate Metabolism and Flavor

Most LAB possess genes related to sugar metabolism and can produce ATP by means of substrate-level phosphorylation of various carbohydrates during fermentation. By determining the genome of *Lactobacillus fermentum* 222 using 454 high-throughput sequencing technology, Illeghems et al. [62] found that this strain possesses different carbohydrate transporter systems, including the phosphoenolpyruvate-dependent sugar phosphotransferase systems (PEP-PTS). Zheng et al. [63] found that aldolase and phosphofrutokinase were more abundant in homozygous *Lactobacillus*, while alcohol dehydrogenase and mannitol dehydrogenase were more abundant in the heterozygous fermentative *Lactobacillus* species. The acid produced by LAB in the metabolic process not only lowered the pH value, inhibited the growth of harmful microorganisms, reduced the water activity (Aw), and extended the shelf-life of the food, but also produced unique flavor substances [8].

#### Metabolism of Glycogen and the Production of Flavor Substances in Meat Products by LAB

Glycogen accounts for approximately 1–2% of the total weight in the muscles, which is a relatively small amount. After the animal is slaughtered, the biochemical reactions in the body gradually stop, and the muscle glycogen is no longer transported through the blood to reach the liver for hepatic glycogen and glucose production. Generally, in the process of fermenting meat products, in order to provide a carbon source for LAB, glucose is added as the energy source.

The specific pathways for the metabolism of sugars and the production of flavor substances by LAB are shown in Figure 3. During the transition from muscle to edible meat after the animal is slaughtered, lactic acid is produced from muscle glycogen during the glycolytic reaction. LAB in the skeletal muscles use glucose to convert to pyruvate during the fermentation process, and pyruvate is produced through the action of various enzymes such as lactate dehydrogenase, acetolactate dehydrogenase, and diacetyl synthase to produce esters with aromas such as acetate and formate and substances with a special flavor such as lactic acid, ethyl coupling, diacetyl, and acetic acid; acetic acid can also be used as a precursor substance for the flavor ester ethyl acetate [64,65,66]. The metabolism of glucose through LAB is divided into homo- and hetero-lactic fermentation, and the fermentation process produces lactic acid, ethanol, and other substances that can lower the pH in the fermentation environment; thus, several LAB can respond to acid stress by regulating their metabolism. Fernandez et al. [67] found that in *Lactobacillus bulgaricus* under acid stress conditions, the pyruvate metabolic pathway could alter to produce fatty acids, which in turn act as the source of flavor substances and add characteristic flavor substances to the foods.

## 3. Application of LAB in the Fermented Meat Products

LAB are used in fermented meat products with functions such as antibacterial and antioxidant effects and nitrite degradation, and the main mechanism is shown in Figure 4. During the production of meat products to produce attractive colors, nitrite is usually added to protect the color, although there is a risk of cancer when the nitrite content is too high. The addition of LAB to meat products can reduce nitrite levels. During the storage period, meat products showcase a common fat oxidation and protein oxidation reaction; appropriate oxidation can add a special flavor to the product, although excessive oxidation produces unpleasant color and odor, resulting in the loss of nutritional value, shortening the shelf-life, and even producing toxic compounds [68]. LAB can slow down the oxidation of meat products through different antioxidant systems. In addition, LAB can produce antibacterial substances such as bacteriocins to prevent the spoilage of meat products and inhibit the growth of pathogenic bacteria.

### 3.1. Nitrite Degradation

It has been shown previously [37] that the addition of *L. plantarum* to meat products can effectively degrade the nitrite content. Microorganisms can use nitrate as an N-source nutrient to conduct assimilative nitrate reduction to nitrite, NO, and N_2_O up to N_2_. Although nitrite offers functions such as color protection and antibacterial and antioxidant properties, as well as prolonging the shelf life of food [69], when its content in food is excessive, it reacts with amino acids to degrade and produce the carcinogenic substance nitrosamine (R_2_N-NO). In recent years, it has been shown that LAB can degrade nitrite in four main ways: through nitrite reductase reducing the nitrite content, through reducing the biogenic amine content, through blocking the formation of N-nitrosamines, and through directly degrading N-nitrosamines [70,71,72]. Sun et al. [73] showed that the addition of *Lactobacillus pentosus* and *Lactobacillus curvatus* to meat products can effectively degrade the N-nitroso content. The inhibitory effect of LAB on N-nitrosamines is not only due to the breakdown of N-nitrosamines through their own enzyme system, but also producing acids that reduce pH to reduce the nitrites and the residue N-nitrosamines produced by them during fermentation. Guo et al. [74] isolated three strains of bacteria from black tea and degraded nitrite at 93% and above. Huang et al. [69] found that, when a variety of LAB were mixed and applied to kimchi fermentation, they not only degraded the nitrite but also produced more flavoring substances such as alcohols, ketones, and olefins during the fermentation. The author showed that LAB could effectively degrade nitrites and reduce nitrite production.

### 3.2. Antioxidant Properties of LAB

During the fermentation of food products, LAB produce small molecules of bioactive peptides and other components, which can exert antioxidant functions through the Keap1-Nrf2-ARE signaling pathway, as shown in Figure 5. In the normal state, Nrf2 in the cytoplasm is bound to its inhibitory protein Keap1. When stimulated through oxidation, the cysteine residues of the Keap1 protein are modified and Nrf2 passes through Keap1 before Nrf2 enters the nucleus, Nrf2 heterodimerizes with muscloaponeurotic fibrosarcoma protein (Maf) on CREB binding proteins (CREB) and binds to antioxidant response elements on DNA (ARE), inducing the expression of various antioxidant enzyme genes such as superoxide dismutase (SOD) and NADPH quinone dehydrogenase. When the cell returns to its normal physiological state, SIRT1 in the nucleus inhibits the Nrf2 expression, allowing Nrf2 to revert to the cytoplasm and maintain normal levels of Nrf2 through ubiquitination (Ub) degradation or negative feedback regulation [75,76]. In addition, protein kinase C (PKC) and extracellular signa-regulated kinase (ERK) are also involved in the regulation of the Nrf2 transcriptional activity by inducing Nrf2 phosphorylation.

LAB, as natural antioxidants, have five regulatory systems: the redox regulatory system, the regulatory signaling pathway, scavenging reactive oxygen species, chelating metal ions, and producing antioxidant molecules; each regulatory mechanism has interrelated regulatory mechanisms [78]. Feng et al. [79] found that *L. plantarum* NJ107, KM119, and *Lactobacillus fermentum* GZ114 isolated from traditional Chinese fermented sausages showed a good antioxidant activity. Ge et al. [80] found that sausages fermented with *Lactobacillus plantarum* NJAU-01 were able to reduce carbonyl production and protect sulfhydryl groups from oxidation. The LAB isolated from donkey milk by Yang et al. [81] not only showed good aroma-producing acidic properties, but could also scavenge hydroxyl (-OH), 1,1-diphenyl-2-picrylhydrazyl, (DPPH) radicals, and superoxide anion radicals (O^2−^). LAB inhibited the auto-oxidation of lipids in fermented meat products, reduced the thiobarbituric acid content, and improved meat quality [80,82]. In conclusion, LAB offer great potential for application as antioxidants of a natural origin when added to food.

### 3.3. Antibacterial Properties of LAB

The mechanism of the antibacterial action of LAB is shown in Figure 4, which mainly plays the role of producing antimicrobial substances, competing with pathogenic bacteria for colonization sites, regulating the flora balance, inhibiting pathogenic bacteria population sensing, regulating cellular immunity, suppressing the inflammatory response, and activating the autophagic response of host cells. LAB, as a type of probiotic, can not only improve the quality of food during the fermentation process, but also inhibit bacteria. LAB inhibit the growth of pathogenic bacteria in the human body by producing organic acids, bacteriocins, hydrogen peroxide, antibacterial peptides, and other substances in the gastrointestinal tract, so that they form a dominant flora and a natural protective barrier in the intestinal mucosa [83]. Mao et al. [84] found that the extracellular supernatant of *L. plantarum* DY-6 could effectively inhibit *Escherichia coli*, *Staphylococcus aureus*, and *Salmonella* with a broad-spectrum bacterial inhibitory effect; the main inhibitory substances were lactic acid, acetic acid, propionic acid, octanoic acid, and capric acid, and the bacterial inhibitory effect was achieved by disrupting the cell membrane structure of the bacteria so that they could not grow and reproduce normally.

LAB are used in meat fermentation to inhibit the growth of acid-sensitive pathogenic bacteria, mainly through acid production. Wang et al. [85] found that when LAB were applied to sausage fermentation, they would proliferate as the dominant flora, resulting in complete inhibition of the growth of pathogenic bacteria, such as *Escherichia coli* in sausages. The bacteriocins produced during the metabolism of LAB also have a positive antibacterial effect, inhibiting the growth and reproduction of undesirable microorganisms such as *Listeria monocytogenes*, *Staphylococcus aureus*, *Escherichia coli*, and *Pseudomonas aeruginosa* [86,87]. LAB are a type of probiotic, and scientists have been screening and characterizing native bacteria that were previously characterized as safe and exhibiting desired metabolic activity for application in fermented meat products, and then using these bacteria to make starter cultures that could provide a standardized flavor and toxicological safety [3].

## 4. Effects of LAB on the Structure of Bacterial Flora in Fermented Meat Products

LAB are usually found at low levels in high-quality raw meat, although the levels increase rapidly through fermentation; in the fermentation process, depending on the raw materials, fermentation environment, fermenting agents, and fermentation process, many kinds of microorganisms are involved; however, in the final product, LAB dominate. The use of strains of a known origin as fermenters can ensure the safety of fermented meat products. Strains of a known origin can not only interact competitively with pathogenic bacteria, but produce organic acid that can also inhibit pathogenic bacterial growth and reduce the use of chemical additives [88]. Du et al. [89] used Illumina technology for the detection of bacterial diversity in fermented meat products to reveal the dominance of the Firmicutes and the phylum Proteobacteria in meat. *Lactobacillus* spp., *Streptococcus* spp., *Carnobacterium* spp., *Lactococcus* spp., and *Leuconostoc* spp. were observed, comprising 21 genera in total, among which *Lactobacillus* spp. LAB are dominant in the fermentation of meat products, and Firmicutes and Proteobacteria are the dominant phyla in fermented meat products. This report was consistent with the results of Tian et al. [90] for analyzing the diversity of microbial communities in naturally fermented air-dried meat, where Proteobacteria accounted for 40%, Firmicutes for 39%, and Bacteroidetes for 14% of the naturally fermented air-dried meat bacterial population.

It has been shown that *L. plantarum* and *Brucella delbrueckii* have a high inhibitory activity against *Clostridium perfringens* during the fermentation of meat products [91]. Table 2 shows some of the recent studies on the LAB in different traditional fermented meat products. In general, the mixture of multiple strains of functional LAB as a concoction of fermenting agents can inhibit the growth of microorganisms and bacteria and reduce the production of harmful substances during meat product fermentation.

## 5. Conclusions and Future Prospects

LAB, as recognized probiotics, possess excellent characteristics such as salt resistance, nitrite resistance, and acid resistance. Fermented meat products are enjoyed by people as a traditional food, which offer good storage and flavor characteristics. However, during storage, they are vulnerable to contamination with miscellaneous environmental bacteria. With the technological advancements in the food industry, people are now adopting artificial inoculation of fermenting agents to ferment meat products. When fermenting meat products, the fermenting agent not only decomposes large molecules of proteins and carbohydrates into small molecules, such as peptides, amino acids, and alcohols that can be easily absorbed by the human body, which gives them unique color and flavor, but it can also produce acids quickly, lowering the pH and Aw of the finished products; inhibit the growth and reproduction of harmful microorganisms; reduce the content of nitrites; and effectively prevent oxidation and rancidity during storage. With the advent of omics science, in the future, researchers will use metagenomics, proteomics, and metabolomics analyses with the sensorial profile of the fermented meat products coming from different regions or areas.

Traditional Chinese fermented meat products have a long history and include several products, such as Jinhua ham, Sichuan sausage, Hunan bacon, and air-dried meat from Inner Mongolia, which are popular because of their long shelf-life and special flavor. Presently, there are some shortcomings in traditional fermented meat products, such as an unclear source of strains, unverified product safety, and the unstable flavor of fermented products, as these factors vary with different regions and production time. With the development of the food industry, the use of LAB, which are beneficial to the human body and can produce fragrances, as a fermenting agent to standardize the production of fermented meat products could significantly improve the food safety of fermented meat products and the stability of the products. LAB, as recognized probiotics, are widely available and easily accessible in China’s vast geographical area, where microbial resources are abundant. However, the fermentation of most conventional fermented meat products relies on the role of microorganisms in the natural environment, which has been inadequately researched, especially in terms of special fermenters. At the same time, research on the functional properties of LAB is at the stage of in vitro experiments. After a series of studies on food processing and storage, whether the food retains its original functional properties or whether its functionality changes after various biochemical reactions in the human body after consumption warrants further research.

In recent years, fermented meat products have received increasing attention in China. As a natural strain of bacteria with a variety of probiotic functions, LAB produce various metabolites during the fermentation of meat products through the processes of fat oxidation, protein hydrolysis, and glycogenolysis, which can interact with each other to form various esters, alcohols, and other substances, improving the quality of fermented meat products. With the improvement in living standards, consumer demands for functional fermented meat products have also increased. Applying functional LAB in meat product fermenters to improve the flavor of the original product and adding certain health benefits to it can usher China’s meat products industry into a new stage of development.

## Figures and Tables

**Figure 1 foods-11-02090-f001:**
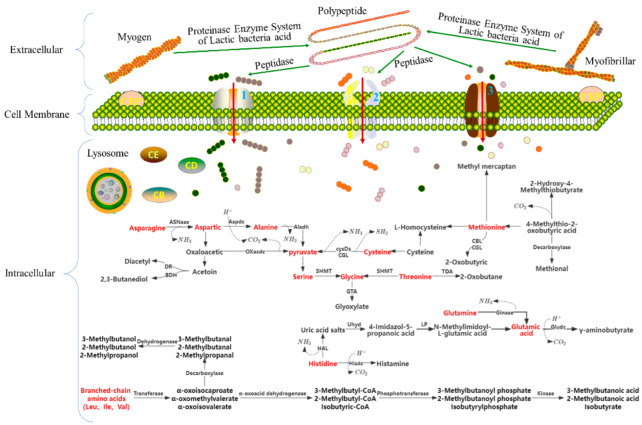
Metabolism of fermented meat products through lactic acid bacteria and the flavor of the substances produced. 1: Oligopeptide transport channels; 2: Small peptide transport channels; 3: Amino acid transport channel; CEP: cell-envelope proteinase; ASNase: asparaginase; Aspdc: ASP decarboxylase; Aladh: alanine dehydrogenase; Oxacdc: oxaloacetate decarboxylase; DR: diacetyl reductase; BDH: 2,3-butanediol Dehydrogenase; CGL: cystathionine-γ-lyase; CBL: cystathionine-β-synthase; SHMT: serine hydroxymethyl transferase; GTA: glycine aminotransferase; TDA: threonine deaminase; Hisdc: histidine decarboxylase; HAL: histidine ammonia-lyase; Uhyd: urocanate hydraatase; IP: imidazolonepropionase; Glnase: glutaminase; Gludc: Glu decarboxylase.

**Figure 2 foods-11-02090-f002:**
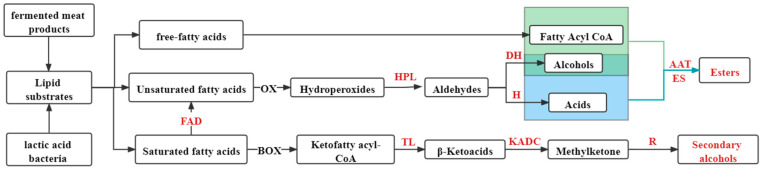
Metabolic pathways of lipids by lactic acid bacteria and the produced flavor substances. Notes: FAD: fatty acid desaturase; HPL: hydroperoxide lyase; TL: thiolase; DH: dehydrogenase; H: hydrogenase; KADC: β-ketoacyldecarboxylase; R: reductase; AAT: alcohol acyltransferase; ES: esterase; OX: oxidation; BOX: β-oxidation.

**Figure 3 foods-11-02090-f003:**
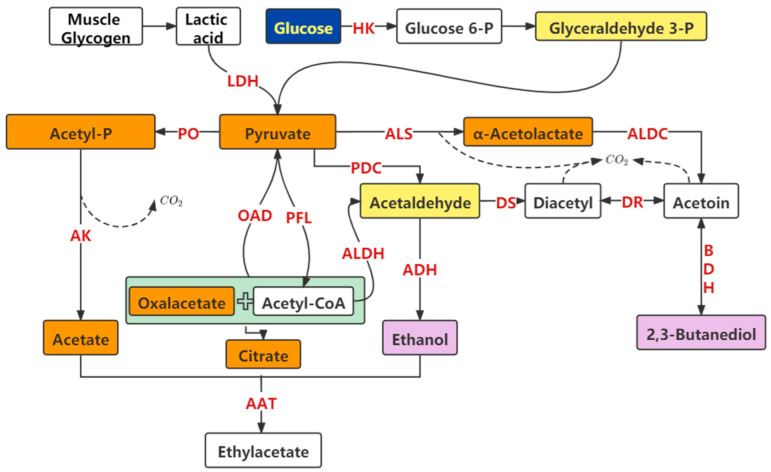
Metabolism of sugar substances by lactic acid bacteria and production of flavor substances. HK: hexokinase; LDH: lactate dehydrogenase; PO: pyruvate oxidase; ALS: acetolactate synthase; ALDC: acetolactate Decarboxylase; PDC: pyruvate decarboxylase; AK: acetate kinase; OAD: oxaloacetate decarboxylase; PFL: pyruvate formate lyase; ALDH: acetaldehyde dehydrogenase; ADH: alcohol dehydrogenase; DS: diacetyl synthase; DR: diacetyl reductase; BDH: 2,3-butanediol dehydrogenase; AAT: alcohol acyltransferase.

**Figure 4 foods-11-02090-f004:**
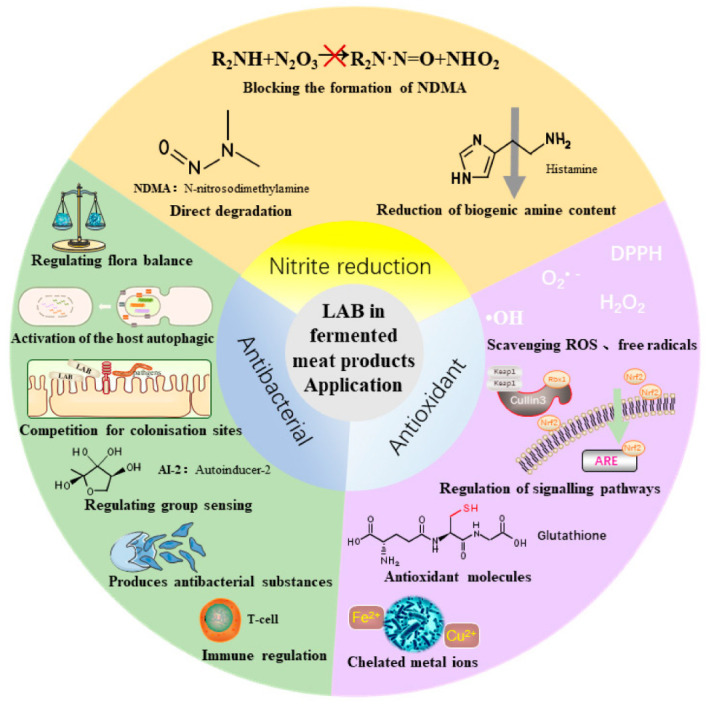
Functional properties of lactic acid bacteria in fermented meat products.

**Figure 5 foods-11-02090-f005:**
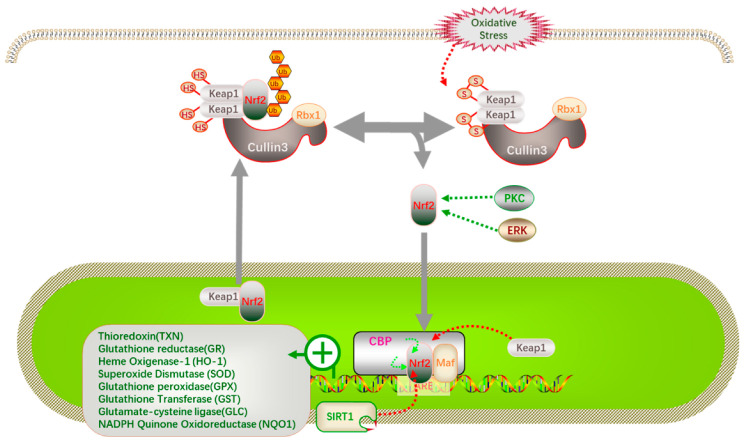
Antioxidant pathways [75,76,77].

**Table 1 foods-11-02090-t001:** Some LAB strains with a cholesterol-lowering ability.

Lactic Acid Bacteria	Source	Strains	Cholesterol Reduction Rate	References
*Lactobacillus rhamnosus*	Human; korean fermented soya beans	LV108; JDFM6	51.93%; 79.00%	[50,51]
*Lactobacillus fermentum*	Human; Tunisian camel milk	grx08; CABA16	38.79%; 58.00%	[51,52]
*Lactobacillus casei*	Human	grx12	38.2%	[51]
*Lactobacillus plantarum*	Fermented Tibetan yak milk; fermented cream in Inner Mongolia	Lp3; KLDS 1.0344	73.30%; 54.08%	[53,54]
*Human Staphylococcus*	Traditional enzymes	M-16	47.41%	[55]
*Enterococcus lumbricoides*	Hainan big fragrant mango	MPL1	57.62%	[56]
*Pediococcus pentosaceus*	Traditional fermented foods in Guizhou Province	MT-4	25.66	[57]
*Lactobacillus kefir*	Indonesian kefir grains	JK17	68.75%	[58]

**Table 2 foods-11-02090-t002:** LAB from different fermented meat products at different stages.

Category	Strains	References
Traditional Taiwanese naturally fermented ham	*Lactobacillus fuchuensis*, *Lactobacillus sakei*, *Lactococcus lactis subsp. Cremoris*, *Lactococcus lactis*, *Lactococcus garvieae*, *Lactococcus lactis subsp. Lactis*, *Leuconostoc. Mesenteroides*, *Leuconostoc citreum*, *Leuconostoc carnosum*, *Enterococcus faecium*, *Enterococcus faecalis*	[92]
Pancetta	*Lactobacillus sakei*, *Lactobacillus curvatus*, *Loigo lactobacillus coryniformis subsp.torquens*	[93]
Prosciutto	*Loigo lactobacillus coryniformis subsp.torquens*, *P. acidilactici*, *Lactobacillus curvatus*, *Lactobacillus plantarum*, *Lactobacillus sakei*	[93]
Jinhua Ham	*Staphylococcus equorum*, *Staphylococcus lugdunensis*, *Lactobacillus Alimentarius*, *Diplococcus lurea*, *Pediococcus pentosaceus*, *Micrococcus mutans*	[94]
Harbin dry sausage	*Pediococcus pentosaceus*, *Lactobacillus brevis*, *Lactobacillus curvatus*, *lactobacillus fermenti*, *Staphylococcus xylosus*, *Lactobacillus sakei*, *Weissella* *hellenica*, *Leuconostoc citreum*, *Lactococcus raffinolactis* and *Lactobacillus plantarum*	[95,96]
Sichuan sausages	*Lactobacillus* spp., *Weissella* spp., *Pediococcus* spp.	[97]
Panxian Ham	*Staphylococcus*, *Micrococcus*	[98]
South Africa sausages	*Enterococcus*, *Staphylococcus*	[99]
Naples-type salami	*Lactobacillus Alimentarius*, *Lactobacillus sakei*, *Staphylococcus*, *Lactobacillus curvatus*, *Staphylococcus xylosus*, *Lactobacillus casei*	[100]

## Data Availability

Not applicable.

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
