# Peer review of "Research Update on the Impact of Lactic Acid Bacteria on the Substance Metabolism, Flavor, and Quality Characteristics of Fermented Meat Products"

_foods, 2022, doi:10.3390/foods11142090_

Round 1

Reviewer 1 Report

Dear, regarding to manuscript submitted for review, my opinion/impression is excellent.
The covered topic is interesting for readers and therefore could be cited.
Manuscript is structuraly well organized, readable, does not require effort to read and understand.
Authors has provided a lot of data related to impact of LAC on the safety and quality of fermented meat products. Also, a lot of graphic has been inserted which make manuscript interesting from visuelization point of view. Cited articles in References are appropriate.

In submitted manuscript, authors tried to explain impact of LAB on the safety and quality of meat products. Despite lack of novelty and originality, the topic is relevant and interesting with scientific and technological point of view, due to impacts of LAB has been explained on more comprehensive way. It could be advantage for reading and citations in future.
Manuscript, including section and subsection are structurally well organised, in logical order, readable, does not require effort to read and understand.  All of the aspects in section and were further well explained and supported by graphics which make manuscript interesting from visualisation point of view.
Cited articles in References are appropriate and recently published.
Conclusions are consistent with the evidence and arguments which has been presented in sections. 

Author Response

Dear, thank you for motivated comments.

Reviewer 2 Report

This review is about the effects of lactic acid bacteria on characteristic of fermented meat products. This is an interesting area of study for many researchers around the world. Overall, the review is well organized and comprehensive.

Comments:

Introduction is too short please add more detail to the introduction

Line 34: add a reference

Line 34: LAB improve (please correct all of the verbs in the manuscript)

Line 56: The style of writing the names of authors in the manuscript is not correct. You should not add a “.” After the names of authors. Please correct the style of references in the manuscript.

Line 61: Maillard reaction

Line 69: molecules

Line 102: add a reference

Line 106: L. plantarum (Italic)

Line 200: This section does not have any references

Line 211: 84.00% is not reported in Table 1

Line 230: delete the space between “expression to”

Line 298: Figure 4.

Line 336: Figure 5.

Line 397: Table 2 is not cited in the manuscript

Reviewer 3 Report

The review article entitle as “ “Research update on the impact of lactic acid bacteria on substance metabolism, flavor, and quality characteristic of fermented meat products”  has some good information regarding fermentation of meat products. The grammar and long sentences are the primary problems of this MS.   

Some of the specific comments are as follows

ABSTRACT

  1. LINE 23-25 ; Since, It is a Review ..Hence these sentences should go at first
  2. LINE 62 : What is Merad reaction ?

3.      Line 378, Is there any studies have shown that eating fermented meat products which of the origin strain of LAB harmful to people's health?

4.      Line 415, Explain how to verify the product stability and safe strain inoculation for fermented meat products coming from different regions or areas.

5.      Line 201-232, there should be more in-text citations from authors.

General comments:

  • Abstract may be modified for better understanding
  • Use short sentences with clarity and satisfying grammar.
  • The typical flavor of the fermented product is the USP of the these product but same time, people who are not used to be these cannot eat these products. So what should be done for making them widely accepted ?
  • Many places, Citation of ref. is not as per journal style

Reviewer 4 Report

This manuscript is a review concerning the impact of lactic acid bacteria on metabolism, flavor and quality characteristic of fermented meat products. It is well written and presented. 

I have some remarks :

line 41-42 add reference 

species (app.) Should not be in italic for all bacteria in the text 

Please check the write nomenclature of all the strains cited in the text 

Line 56 : BP2 should not be in italic 
